# Efficacy of Tango Argentino for Cancer-Associated Fatigue and Quality of Life in Breast Cancer Survivors: A Randomized Controlled Trial

**DOI:** 10.3390/cancers15112920

**Published:** 2023-05-26

**Authors:** Friedemann Schad, Thomas Rieser, Sarah Becker, Jessica Groß, Harald Matthes, Shiao Li Oei, Anja Thronicke

**Affiliations:** 1Research Institute Havelhöhe, Hospital Gemeinschaftskrankenhaus Havelhöhe, 14089 Berlin, Germany; 2Interdisciplinary Oncology and Palliative Care, Hospital Gemeinschaftskrankenhaus Havelhöhe, 14089 Berlin, Germany; 3Institute for Social Medicine, Epidemiology and Health Economics, Charité—Universitätsmedizin Berlin, Corporate Member of Freie Universität Berlin, Humboldt-Universität zu Berlin, and Berlin Institute of Health, 10117 Berlin, Germany; 4Breast Cancer Centre, Hospital Gemeinschaftskrankenhaus Havelhöhe, 14089 Berlin, Germany; 5Institute for Gastroenterology, Hospital Gemeinschaftskrankenhaus Havelhöhe, 14089 Berlin, Germany

**Keywords:** breast cancer, dance, integrative oncology, fatigue, physical activities, quality of life, supportive care, tango Argentino

## Abstract

**Simple Summary:**

Persistent impairments in quality of life, particularly in cancer-associated fatigue, are a major limitation for breast cancer survivors. As physical activity and mindfulness interventions have been shown to be effective, the question arose whether a supervised Argentine tango program may also reduce fatigue symptoms. Evaluation of the self-reported quality of life parameters revealed that quality of life, including fatigue, improved after six weekly one-hour tango sessions. The TANGO trial shows that an Argentine tango program may be an efficacious intervention to improve fatigue and quality of life in breast cancer survivors.

**Abstract:**

Background: Persistent impairments of quality of life—in particular, cancer-associated fatigue—are a major limitation for breast cancer survivors. As physical activity and mindfulness interventions have been shown to be effective in reducing fatigue symptoms, we investigated the efficacy of a six-week Argentine tango program. Methods: A randomized controlled trial was conducted with 60 breast cancer survivors diagnosed with stage I-III tumors 12–48 months prior to study enrollment and who had increased symptoms of fatigue. The participants were randomly assigned with a 1:1 allocation to either the tango or the waiting group. The treatment consisted of six weeks of supervised weekly one-hour tango group-sessions. Self-reported fatigue and further quality of life parameters were assessed at baseline and six weeks post-baseline. Longitudinal changes, correlations, Cohen’s D (*d*) effect sizes, and association factors were also calculated. Results: Superiority of the tango intervention over the waiting list control was found in terms of improvement in fatigue (*d* = −0.64; 95%CI, −1.2 to −0.08; *p* = 0.03), especially cognitive fatigue. In addition, a superiority of the tango intervention over the waiting list was found in the improvement of diarrhea (*d* = −0.69; 95%CI, −1.25 to −0.13; *p* = 0.02). A pooled pre-post analysis of the 50 participants completing the six-week tango program revealed a close to 10% improvement of fatigue (*p* = 0.0003), insomnia (*p* = 0.008) and further quality of life outcomes. Adjusted multivariate linear regression analyses revealed the greatest improvements for participants who were more active in sports. In particular, survivors who received endocrine therapies, were obese, or had no prior dance experience seemed to especially benefit from the tango program. Conclusions: This randomized controlled trial demonstrated that a six-week Argentine tango program improves fatigue in breast cancer survivors. Further trials are warranted to determine whether such improvements lead to better long-term clinical outcomes. Trial registration: trial registration number DRKS00021601. Retrospectively registered on 21 August 2020.

## 1. Introduction

Breast cancer remains the most common cancer in women worldwide [1], and approximately one third of these women experience moderate to severe symptoms of fatigue [2,3]. Cancer-related fatigue is a multidimensional syndrome, having a profound negative impact on health-related quality of life (QoL). Fatigue symptoms are most pronounced during cancer treatment and often persist for many years afterwards [3,4,5,6]. In systematic reviews, it has been concluded that benefits for QoL can be reached with physical activity and interventions such as yoga, exercise, physical self-management, and cognitive behavioral therapies [7,8]. Currently, it is widely accepted that exercise and physical activities be proposed as an add-on first-line intervention [9]. A meta-analysis revealed that a supervised combination of resistance training and aerobic training is the most effective physical exercise for reducing fatigue in breast cancer patients [10]. In addition, mindfulness-based interventions, such as meditation and yoga in particular, have been shown to be efficacious to reduce fatigue levels [11]. Another suitable treatment for improving psychological and physical outcomes in cancer patients is music therapy; in fact, it has been found to have positive effects on anxiety, fatigue, and QoL [12]. As dance utilizes a combination of physical activity, music, and mindful elements, it might be an appropriate and effective approach to address fatigue symptoms and to improve QoL [13]. Systematic reviews have revealed that different art therapies improve anxiety, depression, fatigue, stress, and QoL, and that cancer-related cognitive complaints especially might be managed through multimodal approaches [14,15,16]. Dance can be understood as a multimodal treatment based on body awareness, expression, and rhythm, combining mindfulness and physical activity [17,18]. However, for breast cancer patients, fatigue and lack of motivation, as well as lack of time, are often barriers to engaging in physical activity [19]. Thus, music-based interventions such as dance might provide motivation for increased physical activity [20]. A dance program conducted in five European countries for patients with breast cancer showed positive changes in QoL and improvements in the emotional and the social scales [21]. Argentine tango is assumed to be suitable for reducing fatigue symptoms, as a systematic review on Argentine tango in Parkinson’s disease reported beneficial effects on fatigue and QoL [22]. To examine the effect of this dance style in cancer patients, a feasibility study on Argentine tango for cancer survivors has been performed [23]. However, a randomized trial is needed to measure the efficacy of tango on fatigue symptoms and further QoL outcomes.

The purpose of this clinical trial was to evaluate the efficacy of a six-week Argentine tango program on fatigue in breast cancer survivors. It was expected that fatigue would improve in the tango group compared to the waiting group. In addition, it was anticipated that the six-week tango program would have additional QoL benefits as well.

## 2. Materials and Methods

This clinical trial was an investigator-initiated trial approved by the responsible local ethics committee (Ethik-Kommission der Ärztekammer Berlin, on 2 April 2020 with the reference number Eth-05/20) and has been registered (trial registration number DRKS00021601). The study protocol was published [24]. All eligible participants provided written informed consent for their participation in the study before enrollment, and the trial was reported in compliance with the Consolidated Standards of Reporting Trials (CONSORT) reporting guidelines.

### 2.1. Participants, Study Design, and Procedure

The detailed methods of the TANGO trial have been reported and published in the study protocol [24]. Briefly, the TANGO trial was a single-center, 2-arm, randomized controlled clinical trial conducted at the Research Institute Havelhöhe gGmbH at the Hospital Gemeinschaftskrankenhaus Havelhöhe, Berlin, Germany. The participants were recruited between June 2020 and May 2022 at the primary care facility of the Breast Cancer Centre at the Hospital Gemeinschaftskrankenhaus Havelhöhe. To be included in the study, participants had to be diagnosed with stage I to III breast cancer 12–48 months before enrollment, had to have completed their primary oncological therapy (e.g., surgery, radiotherapy, chemotherapy), and had to have had increased fatigue levels as diagnosed with the German version of the Cancer Fatigue Scale (CFS-D) [25] (≥12 points in the CFS-D questionnaire). All participants, clinicians, and therapists were blinded to the randomization and assignment process, and eligible participants were allocated in a 1:1 ratio to either the tango or the waiting arm. Then, baseline assessments were conducted.

### 2.2. Intervention

The tango program consisted of six weekly one-hour Argentine tango sessions and was conducted by a professional teacher in small groups of three to eight participants. The tango sessions were conducted in alternating and rotating partner exercises so that all participants learned both the leading and the following roles. The tango exercises were oriented towards walking to music, promoting self-awareness, spatial awareness, and interaction. In brief, the six weekly structured group sessions were comprised of breathing and relaxation exercises, balance finding, tango walking, and dancing elements as well as social interaction. Participants randomized to the waiting arm continued their routine activities during the six-week waiting period and subsequently received the same Argentine tango program as the participants in the tango group.

Adapting to the then prevailing COVID-19 pandemic conditions, the tango sessions were conducted in compliance with the national security and hygienic regulations at that time. The number of sessions each participant actually attended was recorded. The dance program was led by a tango teacher with extensive experience, and the safety of the participants was warranted. The participants were encouraged to report any adverse events, discomfort, or problems they experienced. Post-exertional symptom exacerbation after exercise may occur [26], which was therefore monitored. In cases of severe mobility, balance, or neuro-motoric deficits, precautions similar to those recommended for tango for persons with Parkinson’s disease [27] would have been taken.

### 2.3. Outcome Measures

Demographic, behavioral, and medical information was collected with the help of clinical interviews, and self-reported fatigue—as well as QoL—was surveyed prior to and after the intervention. The primary outcome measure was the German version of the Cancer Fatigue Scale (CFS-D), consisting of a 15-item questionnaire on 3 subscales (physical, cognitive, and affective fatigue) with a possible range of 0 to 60 (maximum fatigue) [25]. The secondary outcomes included the Pittsburgh Sleep Quality Index (PSQI), which is composed of seven sleep-related subscales (subjective quality, latency, duration, habitual efficiency, disturbances, use of sleeping medication, and daytime sleepiness) and ranges from 0 to 21 (maximally disturbed sleep) [28]. In addition, the self-reported QoL was investigated with the European Organization for Research and Treatment of Cancer Questionnaire C30 (EORTC-QLQ-C30) [29], which is structured into 15 different subscales (1 global health, 5 functional, and 9 symptoms, including fatigue and insomnia scales). We followed the EORTC-QLQ-C30 Scoring Manual for equations, calculation, and the handling of missing data.

### 2.4. Statistical Analysis

All statistical analyses were performed using Excel 2010 (Microsoft, Redmond, WA, USA) and the R software (Version 4.2.1, R core team, Vienna, Austria), a language and environment for statistical computing. One-way ANOVA was performed to determine the between-group mean differences. 95% confidence intervals (CI), Cohen’s D effect sizes (*d*) (*d* > 0.2 were considered as small, *d* > 0.5 as medium, and *d* > 0.8 as large effect sizes), and significance (*p*-values < 0.05 are considered to be significant) were calculated. For pre-postanalysis of the entire tango group, adjusted multivariate linear regression analyses were performed to identify influencing factors and to address potential sources of bias and confounders. Predicting variables (with regard to T0) were age (in years), the respective QoL scale values at T0, the UICC cancer stages (stage I to III; numerical 1–3), the intake of endocrine therapy (no therapy, 0; therapy, 1), participation in sport activities (no sport, 0; occasionally, 1; regularly, 2), possible dance experience (no experience, 0; experience in the past, 1; currently, 2), the menopausal status (pre-, peri-, and postmenopausal), and BMI (normal, 18.5 ≤ BMI < 25; overweight, 25 ≤ BMI < 30, and obese, 30 ≤ BMI).

## 3. Results

Overall, 272 breast cancer survivors were screened and contacted by the study center; 185 were not suitable because they were not interested (distance too far, no time, pandemic situation) or did not meet all the inclusion criteria. In total, 87 interested candidates were referred to the study physician for a clinical interview, and 60 eligible participants were enrolled and randomized—following the randomization sequence—in the temporal order of the study inclusion and were allocated 1:1 to the tango (n = 30) or the waiting (n = 30) group (Figure 1). Of these 60 enrolled participants, the mean age (SD) was 60.5 (10.3) years and the mean time between first diagnosis and study entry was 1.7 (0.7) years. Two patients dropped out of the tango group (appointment scheduling or medical issue), and two patients dropped out of the waiting group (unwillingness to participate; had no time). The groups were equivalent in terms of baseline characteristics, which are summarized in Table 1.

### 3.1. Group Comparisons

In total, 24 participants in the tango group attended at least four tango sessions and were compared to the 28 participants of the waiting group (Figure 1). For the primary outcome of fatigue, superiority was found for the tango intervention (*p* = 0.03) over waiting, with a medium effect size for the CFS-D total scale (*d* = −0.64; 95%CI, −1.2 to −0.08) and especially for the cognitive fatigue subscale (*d* = −0.62; 95%CI, −1.2 to −0.06) (Table 2).

Boxplot analyses illustrate the between-group differences for the CFS-D total and the cognitive fatigue subscale (Figure 2).

Furthermore, a 15% group difference (*p* = 0.02) with a medium effect size was found for the improvement of the symptom diarrhea (*d* = −0.69; 95%CI, −1.25 to −0.13) (Table 2). Additionally, improvements with small effect sizes were observed for further secondary outcomes: the PSQI subscales daytime sleepiness (*d* = −0.41) and sleep latency (*d* = −0.25); the EORTC QLQ-C30 subscales global health status (*d* = 0.48), physical (*d* = 0.48), cognitive (*d* = 0.33), and social (*d* = 0.41) functioning scales; and the symptom loss of appetite (*d* = −0.43) (Table 2).

### 3.2. Pre-Postanalyses

A total of 50 participants completed the six-week tango program and evaluations of the outcome measures were assessable (Figure 1). Pre-postanalyses for the entire cohort revealed improvements for the CFS-D total and all CFS-D subscales and also for the PSQI total and the PSQI daytime sleepiness subscale (Table 3).

Similarly, improvements for several EORTC QLQ-C30 scales were observed (Table 3) to be especially close to or equal to 10% improvements for fatigue (*p* = 0.0003), social functioning (*p* = 0.02), insomnia (*p* = 0.008), and diarrhea (*p* = 0.01) (Figure 3).

To identify association factors, linear multivariate regression analyses were performed for all longitudinal changes in QoL questionnaires considering age, body mass index (BMI), cancer stages, respective baseline scale values at T0, endocrine therapy, sport activities, and prior dance experiences. Table 4 presents a summary of multivariate regression analyses for the changes of the CFS-D total and the CFS-D subscales; for the EORTC QLQ-C30 global health status (QL), physical (PF), emotional (EF), and cognitive (CF) functioning; and for the symptom fatigue (FA). Associations were found between endocrine therapy and the improvement of the CFS-D, in particular the improvement of physical fatigue (estimate β = −7.7; *p* = 0.03) (Table 4, upper panel). In addition, association of an obese BMI with an improved physical fatigue (estimate β = −12.8; *p* = 0.04) and similarly an improved emotional functioning (β = 27.7; *p* = 0.007) were found (Table 4, lower panel). Another finding was the association between enhanced sport activities and improved cognitive functioning (β = 8.4; *p* = 0.006) and fatigue (β = −5.5; *p* = 0.04). On the other hand, another association was found between pre-existing dance experience with a lower improvement of fatigue (β = 13.1; *p* = 0.006) and cognitive functioning (β = −15.5; *p* = 0.003) (Table 4, lower panel).

### 3.3. Safety and Dropout

All participants survived the study period, and none developed recurrence or metastases. For the entire study cohort, no mild, moderate, or severe adverse events were recorded. No case of symptom exacerbation was reported in this trial.

A total of 56 participants received the tango program, and 50 of them (89%) completed the tango lessons (Figure 1). In total, 32 participants (64%) attended all six tango sessions, 14 (5%) attended five sessions, and 4 participants (8%) attended four sessions. Six study participants (11%) attended fewer than four sessions and were excluded from the analysis (Figure 1). Reasons for missing single sessions were the pandemic situation or COVID−19 vaccination, illness as well as family, work, or other obligations. None of the participants dropped out of the tango program due to a lack of interest.

In Table 5 the baseline characteristics of the 50 assessable participants with those who participated in fewer than four tango sessions are compared. Most characteristics were equivalent in terms of age, time since first diagnosis, and tumor biology, with the exception of a higher BMI (*p* = 0.01) and more receipt of chemotherapy (*p* = 0.05) in dropouts (Table 5).

## 4. Discussion

The present randomized controlled trial is to our knowledge the first efficacy study to examine an Argentine tango program for reducing fatigue symptoms in breast cancer survivors. Superiority of the tango intervention over waiting was found for the improvement of fatigue. Medium effect sizes were observed for fatigue, especially cognitive fatigue. Furthermore, pre-postanalyses of the entire tango study cohort revealed a 10% improvement in fatigue, insomnia, and other QoL parameters. Moreover, multivariate regression analyses indicated that a higher level of sport activity was associated with an increase in the effectiveness of this tango program. The presence of obesity or the use of endocrine therapies appeared to be associated with an increased effectiveness of this tango program. Furthermore, the positive effect of the tango intervention was increased for those without previous dance experience. It is assumed that the cognitive gain through the learning experience and novelty may be relatively greater for participants who have never danced before, and thus, the effect sizes obtained for QoL improvements are greater.

Compared to the published effects of physical exercise interventions on fatigue, the effect sizes found here are in accordance to the summarized effects in meta-analyses for supervised interventions for cancer survivors with cancer-related fatigue [30,31]. Additionally, in an earlier randomized trial of breast cancer patients who had insomnia, six weeks of cognitive behavioral therapy resulted in significant benefits, such as a 13% improvement in sleep efficiency and a 10% improvement in global health status [32]. Interestingly, two recently published RCTs also reported QoL benefits resultant of dance interventions for cancer patients, including the improvement of self-reported fatigue [33,34].

Mind–body therapies and physical activity [35] as well as physical self-management interventions [36] have been established to counteract adverse effects of oncological therapeutics, and seem to generate beneficial effects on perceived QoL including fatigue levels [11,37]. Accordingly, some of these interventions including exercises, yoga, relaxation techniques, stress management, and mediation have been included in international guideline recommendations [38,39], and a recent meta-analysis of RCTs revealed that the supervised combination of resistance training and aerobic training is the most effective physical exercise to reduce fatigue symptoms [10]. The results of our multivariate analyses indicate that improvements in fatigue particularly occur among tango participants who exercise regularly. Thus, the generally known supportive effect of physical activity seems also to synergize with this Argentine tango intervention.

Based on current guideline prescriptions, breast cancer survivors are often advised to undergo adjuvant endocrine therapy to prevent cancer recurrence [40], and accordingly, the majority of the study cohort here were taking these medications. A systematic review with meta-analysis found depression to be a predictor for endocrine therapy refusal, which affects many breast cancer survivors [41]. Reduced self-reported cognitive functioning has been shown to be associated with anxiety, depression, fatigue, and menopausal complaints [42]. A recent comprehensive review article reported that cognitive dysfunctions are common side effects of endocrine therapy and that there is a benefit in non-pharmacological therapies, specifically with exercise and cognitive rehabilitation [43]. Interestingly, in the present study, multivariate analyses showed that participants taking endocrine therapies in particular benefited from an improvement in fatigue symptoms. Early discontinuation and non-adherence to adjuvant endocrine therapy are associated with increased mortality in breast cancer survivors [44], and cognitive dysfunctions associated with endocrine therapy are underdiagnosed and undertreated [43]. A tango program as described here could therefore be suitable to support endocrine treatment adherence and thereby further clinical outcomes as well.

Increased fatigue levels and cognitive disorders are common among breast cancer survivors, especially if treated with chemotherapy [45]. It was reported that cognitive decline in breast cancer survivors is mainly due to chemotherapy [46,47]. We have also observed that increased cognitive fatigue particularly occurs in breast cancer patients receiving chemotherapy [48]. There are only few therapeutic options to alleviate cognitive impairments, and it has been reported that Argentine tango can influence and improve cognitive abilities in Parkinson’s disease patients [22]. Argentine tango improves and increases cognitive activity and the participant is constantly learning [27]. Thus, tango may have a positive effect on cognitive fatigue, and thereby on general fatigue symptoms, in breast cancer survivors, as shown in the present trial. It is generally accepted that exercise and physical activity can lower fatigue levels [9]. A music-based, mind–body exercise program such as Argentine tango that is not too elaborate or strenuous and also accommodates the patient’s desire for social interaction [49] might be suitable.

Notably, the most efficacious effects of the tango intervention appeared to be the improvement of diarrhea. Diarrhea is a well-known side effect of many oncological treatments and is experienced by a third of breast cancer patients [50]. Accordingly, fifteen participants (30%) in our study cohort initially self-reported diarrhea (any grade > 1), of whom eleven reported an improvement after the completion of the six-week tango program. That is, Argentine tango could also be helpful in alleviating long-term side effects of oncological therapies, such as diarrhea.

The mechanism by which Argentine tango improves QoL can be explained by the fact that it has a specific effect on improving neurological and psychological functions [22]. In addition, tango improves spatial awareness, which must be stored, remembered, and reused [51]. Thus, in particular, the neurological and cognitive improvements could explain the fact that cognitive fatigue was improved most compared to physical and affective fatigue in our trial. As it requires dancing with a partner, Argentine tango can foster social and personal relationships and support relational goals [22,27]. For our breast cancer survivors, we suggest three additional aspects likely improved their general well-being: (a) the social aspect by fostering teamwork, community involvement, social networking, and non-verbal communication; (b) the low-threshold offer of an interesting, pleasant, and—at the same time—intensive dance therapy; and (c) the self-awareness and embracing of one’s own body, leading to improved self-perception.

Limitations of this study include the use of a waiting but not an additional active waiting group, which precludes the analysis of group effects. An additional limitation is a potential recruitment bias. Furthermore, the recruitment and implementation of the study occurred during the COVID-19 pandemic. Accordingly, the tango sessions had to be conducted under the locally prescribed hygiene conditions and protective regulations.

The strengths of our trial are the randomized controlled trial design, high adherence to the intervention despite the COVID-19 pandemic situation, and an adjusted multivariate regression analysis—reducing potential confounding biases and identifying positive prognostic variables such as BMI, intake of endocrine therapy, exercise practice, and prior dance experience—which maximizes the generalizability of our findings.

## 5. Conclusions

The present randomized controlled trial demonstrated that a six-week Argentine tango program improved the fatigue and the quality of life of breast cancer survivors with cancer-related fatigue. The results suggest that this program has the potential to improve fatigue in particular in breast cancer survivors taking endocrine therapy, showing obese BMI, and/or with no dance experience. Larger trials are warranted to substantiate the results of this study and to determine whether this tango program is useful for preventing fatigue and improving long-term clinical outcomes.

## Figures and Tables

**Figure 1 cancers-15-02920-f001:**
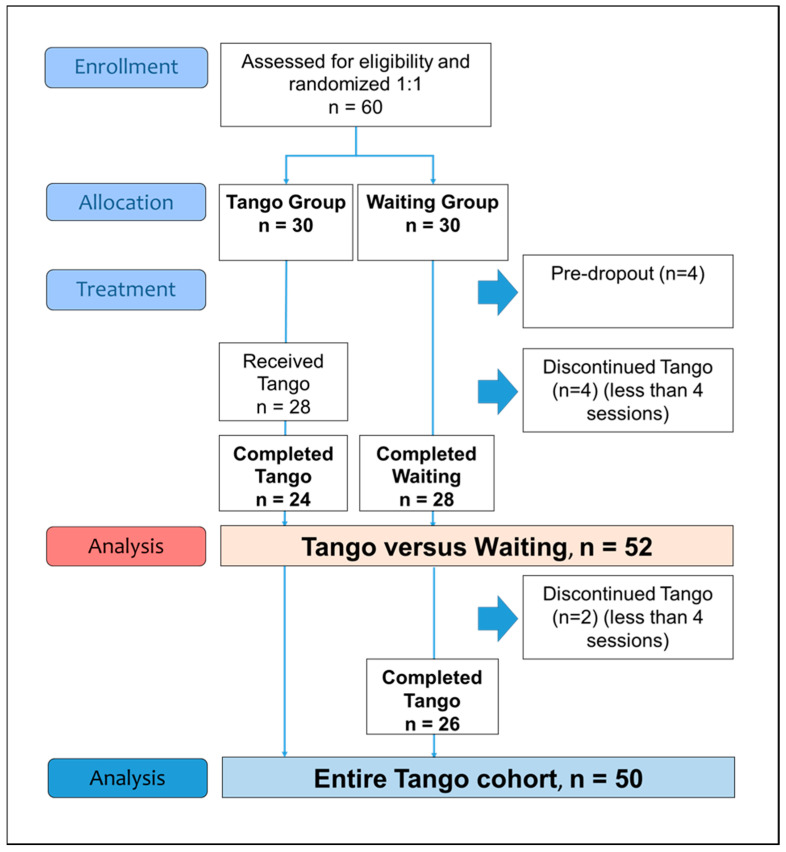
Flow diagram of the study phases.

**Figure 2 cancers-15-02920-f002:**
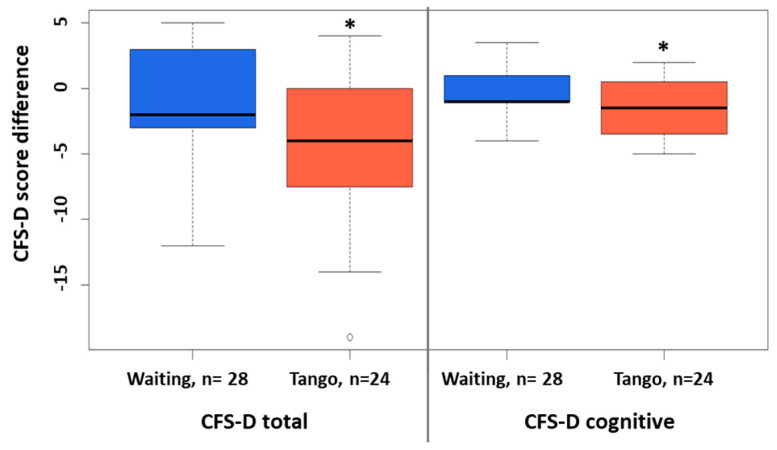
Between-group differences in CFS-D scores tango versus waiting. Boxplots for the between-group differences of the CFS-D scores for six-weeks waiting or tango are shown. Significant differences are indicated. *: *p*-value < 0.05.

**Figure 3 cancers-15-02920-f003:**
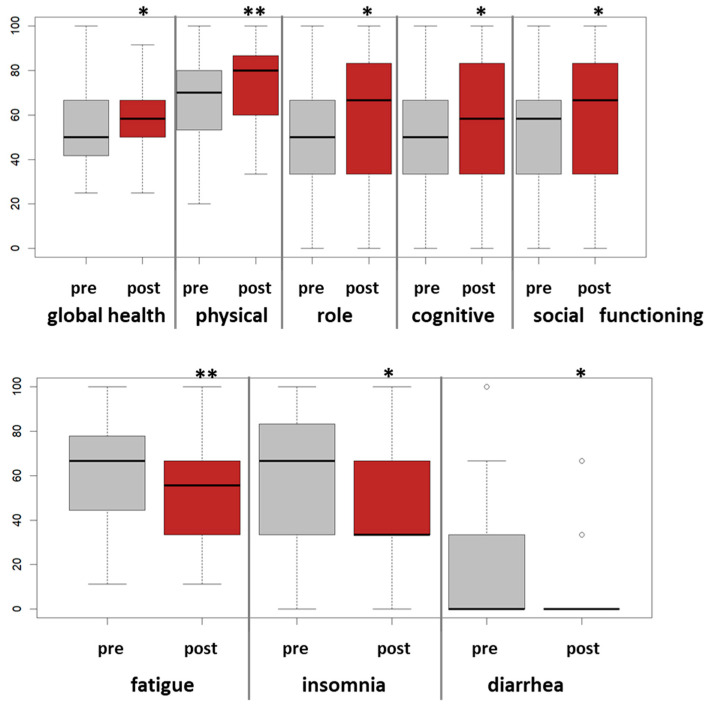
Pre-post analyses of EORTC QLQ-C30 scores for the six-week tango program. For the entire study cohort, n = 50, boxplots for longitudinal changes for the EORTC QLQ-C30 scores pre-post tango are shown. Significant effects are indicated. *: *p*-value < 0.05, **: *p*-value < 0.005.

**Table 1 cancers-15-02920-t001:** Baseline characteristics.

	Total	Waiting	Tango	*p*-Value
**Number of patients**, n (%)	60 (100)	30 (100)	30 (100)	
**Age**, years, mean (SD)	60.5 (10.3)	59.3 (11.0)	61.7 (9.4)	0.377
**Body mass index (BMI)**				
Normal (18.5 ≤ BMI < 25)	39 (65)	21 (70)	18 (60)	
Overweight (25 ≤ BMI < 30)	13 (22)	6 (20)	7 (23)	0.668
Obese (BMI ≥ 30)	8 (13)	3 (10)	5 (17)	
**Currently employed, n (%)**	23 (38)	12 (40)	11 (37)	1
**Living in partnership**, n (%)	39 (65)	19 (63)	20 (67)	1
**Years since first diagnosis:** mean (SD)	1.7 (0.7)	1.7 (0.7)	1.6 (0.6)	0.614
**UICC stages**, n (%)				
I	28 (47)	12 (40)	16 (53)	
II	23 (38)	14 (47)	9 (30)	0.413
III	9 (15)	4 (13)	5 (17)	
**Hormonal status**, n (%)				
Premenopausal	3 (5)	3 (10)	0	
Postmenopausal	57 (95)	27 (90)	30 (100)	0.236
**Triple-negative status, n (%)**	6 (10)	2 (7)	4 (13)	0.667
**Oncological Interventions**, n (%)				
Surgery	60 (100)	30 (100)	30 (100)	1
Radiation	52 (87)	25 (83)	27 (90)	0.704
Endocrine therapy	39 (65)	21 (70)	18 (60)	0.588
Chemotherapy	31 (52)	16 (53)	15 (50)	1
Immunotherapy	6 (10)	3 (10)	3 (10)	1
**Sportive activities**, n (%)				
No	9 (15)	5 (17)	4 (13)	
Occasionally	13 (22)	4 (13)	9 (30)	0.293
Regularly	38 (64)	21 (70)	17 (57)	
**Dance experience**, n (%)				
No	19 (32)	12 (40)	7 (23)	
Yes, in the past	38 (63)	16 (53)	22 (73)	0.273
Yes, current	3 (5)	2 (7)	1 (3)	
**CFS-D**				
Total fatigue score, mean (SD)	32.2 (9.0)	33.7 (7.6)	30.6 (10.0)	0.195
Physical fatigue, mean (SD)	14.1 (4.1)	14.9 (3.8)	13.3 (4.2)	0.109
Affective fatigue, mean (SD)	6.9 (2.6)	7.3 (2.2)	6.4 (2.8)	0.196
Cognitive fatigue, mean (SD)	11.2 (3.5)	11.4 (3.0)	10.9 (4.0)	0.605

CFS-D: Cancer fatigue scale, German version; n: numbers; SD: standard deviation.

**Table 2 cancers-15-02920-t002:** Group differences, six weeks tango versus waiting.

	Tango, n = 24		Waiting, n = 28		Tango vs. Waiting	
	dmean (SD)	*p*-Value	dmean (SD)	*p*-Value	*p*-Value	*d* [95% CI]
**CFS-D-total**	−4.4 (5.4)	**0.001 ***	−1.3 (4.4)	0.14	**0.03 ***	**−0.64 [−1.2; −0.08]**
Physical fatigue	−1.9 (2.5)	**0.002 ***	−0.78 (2.6)	0.14	0.13	**−0.43** [−1.0; 0.12]
Affective fatigue	−1.0 (2.4)	0.052	−0.3 (1.3)	0.32	0.15	**−0.42** [−1.0; 0.14]
Cognitive fatigue	−1.5 (2.2)	**0.004 ***	−0.3 (1.8)	0.44	**0.03 ***	**−0.62 [−1.2; −0.06]**
**PSQI-total**	−0.8 (2.5)	0.16	−0.8 (2.3)	0.08	0.96	0.02 [−0.54; 0.57]
Sleep latency	−0.3 (0.8)	**0.04 ***	−0.1 (0.8)	0.36	0.39	**−0.25** [−0.79; 0.30]
Daytime sleepiness	−0.3 (0.5)	**0.03 ***	−0.04 (0.6)	0.75	0.16	**−0.41** [−0.97; 0.15]
**EORTC QLQ C30**						
Global health	4.7 (16.2)	0.18	−1.9 (11.0)	0.40	0.10	**0.48** [−0.08; 1.03]
Physical functioning	3.3 (12.3)	0.21	−4.1 (17.8)	0.25	0.09	**0.48** [−0.08; 1.03]
Cognitive functioning	6.9 (17.3)	0.07	0.6 (20.7)	0.88	0.25	**0.33** [−0.22; 0.88]
Social functioning	14.9 (27.0)	**0.01 ***	3.6 (28.3)	0.52	0.16	**0.41** [−0.14; 0.96]
Appetite loss	0.0 (16.7)	1	8.3 (21.1)	**0.05 ***	0.13	**−0.43** [−0.98; −0.12]
Diarrhea	−8.3 (19.8)	0.06	7.1 (24.2)	0.14	**0.02 ***	**−0.69 [−1.25; −0.13]**

CFS-D: Cancer fatigue scale, German version; EORTC QLQ C30: European Organization for Research and Treatment of Cancer Questionnaire C30; CI: confidence interval; dmean: mean difference; *d*: Cohen’s D; n: numbers; PSQI: Pittsburgh Sleep Quality Index; SD: standard deviation. Significant *p*-values are indicated: *****: *p*-value < 0.05. Small or medium *d* effect sizes are indicated in bold.

**Table 3 cancers-15-02920-t003:** Pre-postanalyses for six-week tango.

n = 50	dmean	SD	*p*-Value
**CFS-D-total**	−3.6	5.8	0.01 *
Physical fatigue	−1.6	2.8	0.02 *
Affective fatigue	−1.0	2.3	0.06 *
Cognitive fatigue	−1.1	2.3	0.002 *
**PSQI-total**	−0.9	2.7	0.03 *
Daytime sleepiness	−0.2	0.6	0.02 *
**EORTC QLQ-C30**			
Global health status	5.7	15.6	0.02 *
Physical functioning	6.9	14.3	0.001 *
Role functioning	8.2	25.9	0.03 *
Cognitive functioning	6.3	19.7	0.03 *
Social functioning	9.2	27.6	0.02 *
Fatigue	−9.8	17.5	0.0003 *
Insomnia	−10.2	25.6	0.008 *
Diarrhea	−8.7	23.9	0.01 *

CFS-D: Cancer fatigue scale, German version; dmean: mean difference; n: numbers; PSQI: Pittsburgh Sleep Quality Index; EORTC QLQ C30: European Organization for Research and Treatment of Cancer Questionnaire C30; SD: standard deviation. Significant *p*-values are indicated: *: *p*-value < 0.05.

**Table 4 cancers-15-02920-t004:** Association factors for six-week tango and CFS-D and EORTC QLQ C30 changes.

CFS-D, n = 50	Total	Physical Fatigue		Cognitive Fatigue	Affective Fatigue
Adjusted for	age, cancer stage, T0-value, and menopausal status				
**Endocrine therapy**	**−6.825 ***	**−7.725 ***		−6.513	**−7.119 ^+^**
	**Reference BMI normal**				
**BMI overweight**	2.350	0.900		3.159	5.239
**BMI obese**	−7.391	**−12.882 ***		−5.695	−6.478
**Sports**	**−3.711 ***	**−4.732 ***		**−3.760 ^+^**	−3.234
**Dance experience**	**6.781 ***	**6.890 ***		**6.390** ** ^+^ **	**6.930** ** ^+^ **
**EORTC QLQ C30, n = 50**	**QL**	**PF**	**EF**	**CF**	**FA**
Adjusted for	age, cancer stage, T0-value, and menopausal status				
**Endocrine therapy**	−2.847	−1.894	3.242	3.315	−2.367
	**Reference BMI normal**				
**BMI overweight**	−0.940	0.624	5.318	8.703	5.516
**BMI obese**	3.106	**12.775 ^+^**	**27.671 ****	14.755	−6.873
**Sports**	**4.353 ^+^**	0.941	3.609	**8.44 0 ****	**−5.485 ***
**Dance experience**	−5.071	1.091	**−9.575** ** ^+^ **	**−15.511 ****	**13.107 ****

Multivariate linear regression analyses, adjusted for age, UICC cancer stages, the respective CFS-D or EORTC QLQ-C30-values at T0, and menopausal status were performed for CFS-D (upper panel) and EORTC QLQ-C30 (lower panel) scale changes for the six-week tango module. The estimates of the CFS-D scale and subscales were converted to percentages. Improvements of functional EORTC QLQ-C30 scales are indicated by positive whereas improvement of fatigue by negative estimates. Significant *p*-values are indicated: ******: *p*-value < 0.005; *****: *p*-value < 0.05; **^+^**: *p*-value < 0.01. Significant improvements are highlighted in green and deteriorations in red. Nonsignificant improvements with estimates > 5 are highlighted in light green and deteriorations in orange. CFS-D: Cancer fatigue scale, German version; EORTC QLQ C30: European Organization for Research and Treatment of Cancer Questionnaire C30; n: numbers; QL: global health status; PF: physical; EF: emotional; CF: cognitive; FA: fatigue symptoms.

**Table 5 cancers-15-02920-t005:** Characteristics of participants.

n = 56	Completed Tango (89%)	Discontinued Tango (11%)	*p*-Value
Number of patients, n (%)	50 (100)	6 (100)	
Age, years, mean (SD)	61.3 (9.9)	60.5 (12.2)	0.855
**Body mass index (BMI)**			
Normal (18.5 ≤ BMI < 25)	36 (72)	2 (33)	
Overweight (25 ≤ BMI < 30)	10 (20)	1 (17)	**0.012 ***
Obese (BMI ≥ 30)	4 (8)	3 (50)	
**Living in partnership, n (%)**	31 (62)	5 (83)	0.562
**Years since first diagnosis: mean (SD)**	1.7 (0.7)	1.4 (0.1)	0.328
**UICC stages, n (%)**			
I	23 (46)	2 (33)	
II	20 (40)	2 (33)	0.473
III	7 (14)	2 (33)	
**Hormonal status, n (%)**			
Premenopausal	2 (4)	0	
Postmenopausal	48 (96)	6 (100)	1
**Triple-negative status, n (%)**	4 (8)	2 (33)	0.231
**Interventions, n (%)**			
Surgery	50 (100)	6 (100)	1
Radiation	46 (92)	5 (83)	1
Endocrine therapy	34 (68)	4 (67)	1
Chemotherapy	24 (48)	6 (100)	0.048 *
Immunotherapy	6 (12)	0	0.841
**Sport activities, n (%)**			
No	8 (16)	1 (17)	
Occasionally	11 (22)	1 (17)	0.955
Regularly	31 (62)	4 (67)	
**Dance experience, n (%)**			
No	16 (32)	1 (17)	
Yes, in the past	32 (64)	4 (67)	0.363
Yes, current	2 (4)	1 (17)	
**QoL assessment at baseline**			
CFS-D total, mean (SD)	31.01 (9.26)	37.00 (5.23)	0.133
PSQI total, mean (SD)	9.55 (3.43)	11.50 (4.99)	0.229
EORTC QLQ C30 Global health, mean (SD)	54.42 (16.84)	51.67 (20.00)	0.739

CFS-D: Cancer fatigue scale, German version; n: numbers; PSQI: Pittsburgh Sleep Quality Index; EORTC QLQ C30: European Organization for Research and Treatment of Cancer Questionnaire C30; SD: standard deviation. Significant *p*-values are indicated: *: *p*-value < 0.05.

## Data Availability

The datasets that support the findings in this article are not publicly available for privacy and security reasons but can be obtained from the corresponding authors upon reasonable request.

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
