# Peer review of "Efficacy of Tango Argentino for Cancer-Associated Fatigue and Quality of Life in Breast Cancer Survivors: A Randomized Controlled Trial"

_cancers, 2023, doi:10.3390/cancers15112920_

Round 1
Reviewer 1 Report
This is an excellent study of non-pharmacologic intervention for cancer-related fatigue and other aspects of quality of life. I suggest edits around presentation of methodological details and results below. While these are important edits to make, especially regarding removal of references to "clinical significance" without referencing the MCID literature, I consider them pretty easy edits to make and congratulate the authors on a very nice conceptualization, execution, and presentation of work in an important area.
Intro
Line 81: “It was expected that the tango group will have a clinically relevant reduction of fatigue symptoms compared to the waiting group.”
The authors hypothesize that the effect will be clinically relevant but do not define what that means or how they intend to assess. Please either remove references to clinically relevant/significant/important/meaningful or define what this means in terms of published minimum clinically important differences (MCID) for this variable and population. In my opinion, the results are interesting without the verbiage around being clinically relevant so it would be sufficient to make the minor edit of removing that verbiage throughout.
Line 169: Please reframe non-significant findings to clarify that no difference was detected. For instance, if statistics showed no difference in baseline severity between groups, then you would report that groups were equivalent in terms of baseline. If you feel like the p value is close enough to your significance criteria (<0.05) that feel compelled to provide a bit more transparency, you could report p values in an extra column on right side of Table 1.
“group reported non-significant more severe symptoms of fatigue compared to the tango”.
Methods
Please specify acronyms for names when first introduced. For instance, I think GKHB in line 97 was defined as Gemeinschaftskrankenhaus Havelhöhe, Berlin in Line 95. Please amend Line 95 to read “Gemeinschaftskrankenhaus Havelhöhe, Berlin, Germany (GKHB)”
Do you know how long since participants had completed “primary oncological therapy (surgery, radiotherapy, chemotherapy)”? If so, please report.
Please detail the tango intervention a bit more in section 2.2 with regard to the following aspects:
- was tango practiced as partnered? If so, who served as partner: an instructor? companion? volunteer?
- Please specify style of tango practiced: later you specify that you used Argentinian. State that in 2.2 to distinguish between Adapted, Argentinian, and American Ballroom Tango
- What safety precautions were applied, if any. For instance, were recommendations incorporated based on Hackney et al (2010) recommendations for teaching tango to individuals with PD? If no safety precautions were followed beyond those typical of an artistic dance studio, please state so. In this case it would help to report starting characteristics of the cohort in terms of cancer-related neuromotor deficits such as whether there was any evidence of deconditioning or chemotherapy-induced neuropathy (CIN). It’s possible that no safety precautions were strictly necessary if you didn’t have anyone with significant deconditioning or CIN…would just be good to address that explicitly for those seeking to recreate your results.
Hackney, M. E., & Earhart, G. M. (2010). Recommendations for implementing tango classes for persons with Parkinson disease. American Journal of Dance Therapy, 32, 41-52.
Results
1. Please report just the facts. The results are compelling. No need for words such as “substantial”that seem intended to sway the reader’s opinion. For instance:
Line 209 “especially substantial 10% improvements…” should be simplified to “including improvements (10%)…”
2. Please remove use of word “significantly” as a modifier when reporting or discussing findings. When we report improvement, the data are statistically significant by definition, so reusing the word “significant” becomes redundant in way that reads as an opinion. Similarly, use of word “clinically” as modifier should also be avoided unless the report/discussion includes a clear comparison to published minimum clinically important differences (MCID) for the outcome. Here are some instances that require editing below. Please do a search for terms “significant” and “clinically” to find all instances:
Line 208: “Similarly, significant and clinically meaningful improvements for several EORTC QLQ-C30 scales were observed (Table 3), especially substantial 10% improvements…” should be something like “EORTC QLQ-C30 improved, with some sub-scales demonstrating up to 10% improvement in patient-reported symptom severity (Table 3).”
Line 218: “Significant associations.” should be “Associations…”
Line 220: “clinically significant benefits”
Line 225: “a clinically significant association”
3. RE: results of multi-variate regression analysis.
Lines 218-227: “Significant associations were found between receiving endocrine therapy and CFS-D, in particular improvement of physical fatigue (estimate = -7.7; P = .03) (Table 4, upper panel). In addition, clinically significant benefits for obese BMI with respect to normal BMI in improving physical fatigue (esti-221 mate = -12.8; P = .04) were found and similarly for improving emotional functioning ( = 27.7; P = .007) (Table 4, lower panel). Another finding, was a significant association between sport activities and improvements in cognitive functioning ( = 8.4; P = .006) and fatigue ( = -5.5; P = .04). On the other hand, a clinically significant association was found between pre-existing dance experiences associated with lower benefits for fatigue ( = 226 13.1; P = .006) and cognitive functioning ( = -15.5; P = .003) (Table 4, lower panel).”
- Please clarify text describing predictors that increased vs decreased effectiveness of the program. This is interesting information; ideally, take-home messages would be easier to glean. If I’m reading correctly, some of these factors were associated with improved functional gains but prior dance experience seemed to attenuate improvements in fatigue and cognitive functioning. If this interpretation is correct, please clarify throughout results and discussion and consider addressing this finding in your discussion. For instance, you could speculate that novelty of the intervention might have influenced results and/or that prior exposure to the intervention type influenced dose response. It seems to me that your results indicate an impetus to explore whether prior exposure to dance-based training mediates dose response, possibly indicating a need to recommend increased duration or intensity of activity for those with prior dance experience for these participants to optimize functional gains in a way comparable to those who are naïve to the experience.
Discussion
1. Please clarify which predictors increased vs decreased effectiveness of the program in the follow blocks of text:
Lines 272-275: ”Moreover, multivariate regression analyses identified predictors as obesity, the application of endocrine therapy, being more active in sports, or prior dance experience for the effectiveness of the six-week tango program.”
Lines 332-335: “…adjusted multi-variate regression analyses reduced confounding biases and identified positive prognostic variables such as BMI, intake of endocrine therapy, exercises, and also prior dance experience, which maximizes the generalizability of our findings.”
2. RE: Lines 270-272: “Furthermore, pre-post analyses of the entire TANGO study cohort revealed a significantly 10% improvement for fatigue, insomnia, and for further QoL parameters.”
- per earlier comment please remove the term “significantly” as it is redundant. This term restates what is already demonstrated by relevant p-values.
- please clarify. Are you saying here that the moderate effect sizes correspond to a 10% improvement within each of these outcome measures?
3. Lines 337-339: as previously stated, please remove “a significant and clinically relevant”.
Conclusions
1. Suggest rewording this concluding sentence for readability:
“The here presented randomized controlled TANGO trial indicates a significant and 337 clinically relevant improvement of fatigue symptoms and further quality of life aspects 338 with a six-week Argentine tango program for breast cancer survivors.”
Consider editing to something like this: “The TANGO randomized controlled study demonstrated improvements in fatigue and quality of life within 6 weeks of Argentine Tango practice among breast cancer survivors with cancer-related fatigue.”
2. Please define new acronyms used such as FIH
very minor edits suggested in "Suggestions for Authors"
Author Response
Thank you very much for your interest and your very appreciative remarks and comments.
As recommended by you, we have now thoroughly revised and edited the wording and reporting of the significance and have reduced or deleted the terms “clinically”, “significant” and “substantial” throughout the manuscript where appropriate and as suggested by you. The use of acronyms was also corrected accordingly.
The time since study participants completed their primary oncology therapies varied widely and could range from 6 to 18 months after their respective first diagnoses; depending on the therapies they received. Accordingly, the inclusion criteria chosen were between 12 and 48 months after their tumor diagnosis and after the completion of their primary treatments (surgery, radiation and/or chemotherapy). The average time interval between first diagnosis and study initiation for the entire study cohort was 1.7 ± 0.7 years. More detailed information on this can now be found in the Results and Table 1.
The Argentine Tango program was performed in alternating and rotating partner exercises so that participants were able to learn and practice both the leading and following roles. More details on the Tango intervention and safety precautions are given now in the section 2.2. Although none of our study participants had severe neuro-motoric deficits, precautions were taken similar to those described by Hackney & Earhart (2010) [27]. Appropriate additions and further citation have now been made in section 2.2.
Thank you for your attention, indeed, numerous possible conclusions can be driven from the results of our multivariable regression analyses. Presumably, participants also experienced increased fatigue levels from endocrine therapies, which in turn may have been improved by the Tango intervention. Presumably, the cognitive gain is relatively greater for participants who never have had dance experiences before, and thus the effect sizes obtained for improvements of health-related quality of life parameters are greater. Furthermore, the multivariate analyses indicate that improvements in fatigues particularly occur among participants who exercise regularly, thus the generally known supportive effect of physical activity seems here to synergize with this Argentine tango intervention. Some highlights in this regard are now addressed in the discussion.
In the conclusion, your suggestions for reformulation were gladly welcomed and we hope that they meet your approval.
All changes to the revised manuscript are presented in a marked-up version of the document.
Reviewer 2 Report
Very Respected Authors,
What was the total number of participants in the study? How many cancer survivors died in the observed period?
Author Response
Thank you very much for your interest and your remarks.
Sixty breast cancer survivors were enrolled into the study, 52 participants were eligible for group analysis, 50 participants completed the tango program and were eligible for pre-post analysis (Figure 1). All of the enrolled patients survived in the observed period.
Respective additions have now been made in Figure 1 and section 3, all changes to the revised manuscript are presented in a marked-up version of the document.
Reviewer 3 Report
Comments:
1. Title: Its bit of confusing with ‘Tango Argentino’ and ‘TANGO randomized’. Authors may add ‘quality of life’ along with fatigue. There should be a : between Survivors and The TANGO.
2. There are several incomplete or unclear sentences in the Abstract. As it stands alone, authors are advised to mention the sentences with complete meaning.
3. What does it mean ‘superiority for the tango group’ in the Abstract? Please do better comparison.
4. Avoid unnecessary words and long wording sentences.
5. Introduction: What is the association between diarrhea and fatigue or exercise in breast cancer survivors? Can diarrhea lead to fatigue and inflorescence the studied outcome measure.
6. Methods: Study participants’ details are largely missing. Please provide basic information of participants, including total number, age, grouping or intervention and their demographic details.
7. It is not advisable to mention/refer the previously published article [24] for every detail in the Methods section.
8. Line 109 and line 111-112: The information mentioned in these lines is confusing. Mention what participants exactly did in the intervention.
9. Intervention duration is not clear. If it is one hour per week, participants completed only 6 hours of program.
10. Is it possible to provide the exercise intensity details as intervention in this study is a mixed of several items?
11. Please discuss the exercise duration (used in this study) with relevance to the WHO guidelines of exercise recommendation for cancer survivors.
12. Please revise the first sentence in the Discussion section, line 267-268.
13. Authors mentioned “Superiority for the tango group versus the waiting control group was found”. From this statement, it is not clear on which variable such superiority was found.
14. Authors cited several reviews and meta-analyses to support their findings, which is good. However could discuss the possible reasons for the beneficial effects of Tango argentino.
15. Discussion part needs to be further strengthen.
The presentation, structure of sentences and syntax need to be improved.
Author Response
Thank you very much for your interest and your helpful remarks and comments.
- Following your suggestions, we have changed the Title accordingly and we hope you agree:
Efficacy of Tango Argentino on Cancer-Associated Fatigue and Quality of Life in Breast Cancer Survivors: A Randomized Controlled Trial
2.-4. The abstract and other parts of the manuscript now have been thoroughly linguistically revised.
- It is conceivable that symptoms of diarrhea may influence and/or exacerbate fatigue. Surprisingly, the most efficient effects of the Tango intervention seemed to be the improvement of diarrhea, even though at baseline assessment two-thirds of the study participants reported no diarrhea symptoms at all. To date, no direct association between diarrhea and fatigue or exercise has been published for breast cancer survivors. However, this item was not the primary research question for the present study and therefore, this aspect was not explicitly dealt with and thus only presented in a condensed form in the manuscript. Appropriate amendments to address this issue have now been implemented in the discussion.
6.-8. In the sections Methods, Results and in Table 1, the presentation and information on the study participants have now been supplemented and revised.
9.-11. In fact, the Tango program carried out here consists of only six 1-hour weekly sessions, i.e. a total of 6 hours of instruction and dance within 6 weeks, which is considerably less intensive than the time recommended for exercises by the WHO guidelines. But, tango is more than just physical exercise, tango also contains musical and mindful elements, and also motivates physical activity beyond that [20]. However, it is assumed that concomitant physical activities took place alongside the lessons or are continued independently, so that overall, a good increase in physical activity can be achieved in accordance with the general guidelines. And indeed, the multivariate regression analyses indicated that fatigue symptoms improved particularly in participants who exercised regularly. Corresponding notes have now been added in the discussion.
12.-13. The wording of some sentences in the discussion was revised.
14.-15. In the discussion, the possible reasons for the effectiveness of the Tango Intervention were discussed and further references have now been included.
All changes to the revised manuscript are presented in a marked-up version of the document.
Round 2
Reviewer 3 Report
Appreciate authors for revising the manuscript.
However, I cannot find the point-by-point response to each of my comment. Unless the detailed response to each of my comment, I can’t make sure whether my comments were addressed by the authors or not. Please attend all the comments one by one, not combining of 2 or 3 comments and respond simply.
Also please mark or mentioned where (page number of line numbers) where the changes were done in the revised manuscript according to the comments.
It would be good, if authors can polish the language for smooth understanding.
Author Response
Thank you very much for your helpful remarks and comments, now we have answered the queries of the round 1 point-by-point and specify exactly where changes have been made.
Query 1. Title: Its bit of confusing with ‘Tango Argentino’ and ‘TANGO randomized’. Authors may add ‘quality of life’ along with fatigue. There should be a : between Survivors and The TANGO.
Answer to 1. According your suggestions, we have changed the Title which now reads:
Efficacy of Tango Argentino for Cancer-Associated Fatigue and Quality of Life in Breast Cancer Survivors: A Randomized Controlled Trial
Query 2. There are several incomplete or unclear sentences in the Abstract. As it stands alone, authors are advised to mention the sentences with complete meaning.
Answer to 2. The complete Simple Summary and the Abstract now have been thoroughly linguistically revised.
Query 3. What does it mean ‘superiority for the tango group’ in the Abstract? Please do better comparison.
Answer to 3. The sentences including the term “superiority” (lines 33-36) were reworded to better express the meaning.
Query 4. Avoid unnecessary words and long wording sentences.
Answer to 4. Throughout the manuscript, numerous words were deleted and some long sentences were shortened.
Query 5. Introduction: What is the association between diarrhea and fatigue or exercise in breast cancer survivors? Can diarrhea lead to fatigue and inflorescence the studied outcome measure.
Answer to 5. The association between diarrhea and fatigue was not the primary objective of the present study, and no direct association between diarrhea and fatigue or physical activity in breast cancer survivors has been published so far. Since the most efficacious effects of the tango intervention seemed to be the improvement of diarrhea, this item will be dealt in more detail in an upcoming follow-up study of this tango trial and not yet in the introduction of this manuscript. A few amendments in this regard can be found in the discussion lines 331-337.
Query 6. Methods: Study participants’ details are largely missing. Please provide basic information of participants, including total number, age, grouping or intervention and their demographic details.
Answer to 6. All details of the study participants are listed in table 1 (lines 174), some supplemented information has now been added (lines 168-175) and in an extra column in table 1 the p-values are given now.
Query 7. It is not advisable to mention/refer the previously published article [24] for every detail in the Methods section.
Answer to 7. Throughout the manuscript, mentioning and referring to [24] was several times deleted.
Query 8. Line 109 and line 111-112: The information mentioned in these lines is confusing. Mention what participants exactly did in the intervention.
Answer to 8. The description of the intervention now is revised (lines 109-126).
Query 9. Intervention duration is not clear. If it is one hour per week, participants completed only 6 hours of program.
Answer to 9. In fact, the duration of the tango program carried out here consists of only six 1-hour weekly sessions, thus a total of 6 hours.
Query 10. Is it possible to provide the exercise intensity details as intervention in this study is a mixed of several items?
Answer to 10. Each weekly exercise session lasts 60 minutes. In addition to the tango exercises, the usual sports activities are also performed independently, so that the overall intensity of the physical activities performed by the participants varies individually. And indeed, including participants' individual sporting activities as a variable in the multivariate analyses revealed that fatigue symptoms improved, particularly among participants who exercised regularly.
Query 11. Please discuss the exercise duration (used in this study) with relevance to the WHO guidelines of exercise recommendation for cancer survivors.
Answer to 11. Tango is more than just physical activity. Tango also contains musical and mindfulness elements and also motivates physical activity. Therefore, this intervention is discussed here not focusing only on physical activity. In addition to the tango exercises, the usual sports activities are also performed, so that overall a good boost in physical activity can be achieved in accordance with the general guidelines. Corresponding notes have now been added in the discussion lines 273-275 and lines 295-300.
Query 12. Please revise the first sentence in the Discussion section, line 267-268.
Answer to 12. The first sentence of the discussion now has been reworded and now reads:
The present randomized controlled trial is to our knowledge the first efficacy study to examine an Argentine tango program in reducing fatigue symptoms in breast cancer survivors.
Query 13. Authors mentioned “Superiority for the tango group versus the waiting control group was found”. From this statement, it is not clear on which variable such superiority was found.
Answer to 13. The sentence in the discussion (lines 270-271) was reworded to clarify the meaning.
Query 14. Authors cited several reviews and meta-analyses to support their findings, which is good. However could discuss the possible reasons for the beneficial effects of Tango argentino.
Answer to 14. Potential beneficial effects of tango now are discussed in lines 338-349.
Query 15. Discussion part needs to be further strengthen.
Answer to 15. Many passages of the discussion have now been revised and we have taken the help of a native English speaker for another lingual revision of the whole manuscript.
Thank you for your comments.